# Fertility Counseling in *BRCA1/2*-Mutated Women with Breast Cancer and Healthy Individuals

**DOI:** 10.3390/jcm11143996

**Published:** 2022-07-10

**Authors:** Joanna Kufel-Grabowska, Amira Podolak, Daniel Maliszewski, Mikołaj Bartoszkiewicz, Rodryg Ramlau, Krzysztof Lukaszuk

**Affiliations:** 1Department of Oncology, Poznan University of Medical Sciences, 61-701 Poznan, Poland; jkufel-grabowska@ump.edu.pl (J.K.-G.); rodrygramlau@ump.edu.pl (R.R.); 2Department of Obstetrics and Gynecological Nursing, Faculty of Health Sciences, Medical University of Gdansk, 80-210 Gdansk, Poland; amira.podolak@invicta.pl (A.P.); luka@gumed.edu.pl (K.L.); 3Department of General and Oncological Surgery, Wojewódzki Szpital Specjalistyczny im. Janusza Korczaka w Słupsku Sp. z o.o., 76-200 Słupsk, Poland; danielmal2@gmail.com; 4Department of General and Oncological Surgery at Specialist Hospital in Koscierzyn, Sp.z.o.o., 83-400 Kościerzyna, Poland; 5Swissmed Health Center, 80-210 Gdansk, Poland; 6Department of Immunobiology, Poznan University of Medical Sciences, 60-806 Poznan, Poland; 7Invicta Research and Development Center, 81-740 Sopot, Poland

**Keywords:** breast cancer, oncofertility, RRM, RRSO, *BRCA1/2*

## Abstract

Breast cancer is the most commonly diagnosed cancer worldwide and the fifth leading cause of cancer death. In 2020, there were 2.3 million new cases, and 685,000 women died from it. Breast cancer among young women under 40 years of age accounts for 5% to 10% of all cases of this cancer. The greater availability of multi-gene sequence analysis by next-generation sequencing has improved diagnosis and, consequently, the possibility of using appropriate therapeutic approaches in *BRCA1/2* gene mutation carriers. Treatment of young breast cancer patients affects their reproductive potential by reducing ovarian reserve. It can lead to reversible or permanent premature menopause, decreased libido, and other symptoms of sex hormone deficiency. This requires that, in addition to oncological treatment, patients are offered genetic counseling, oncofertility, psychological assistance, and sexological counseling. Given the number of *BRCA1/2* gene mutation carriers among young breast cancer patients, but also thanks to growing public awareness, among their healthy family members planning offspring, the possibility of benefiting from preimplantation testing and performing cancer-risk-reduction procedures: RRM (risk-reducing mastectomy) and RRSO (risk-reducing salpingo-oophorectomy) significantly increase the chance of a genetically burdened person living a healthy life and giving birth to a child not burdened by the parent’s germline mutation. The goal of this paper is to show methods and examples of fertility counselling for *BRCA1/2* gene mutation carriers, including both patients already affected by cancer and healthy individuals.

## 1. Introduction

Breast cancer is the most commonly diagnosed cancer worldwide and the fifth leading cause of cancer death. In 2020, there were 2.3 million new cases, and 685,000 women died from it [1]. Breast cancer among young women under 40 years of age accounts for 5 to 10% of all cases of this cancer. Age is an independent prognostic factor, and young women are more likely to be diagnosed with more aggressive biological subtypes: triple-negative (lacking estrogen and progesterone receptor expression in tumor cell nuclei and without HER2 gene amplification) and HER-2 positive [2]. Survival of young patients with hormone receptor positive carcinomas, mainly luminal B, is worse than that of older women with the same type of cancer. The more aggressive course of this disease may be due to molecular differences of the tumor, despite clinical similarities [3].

The incidence of cancer in this age group is associated with different risk factors than in older women; more often, mutations in genes increasing the risk of cancer are diagnosed, e.g., *BRCA1/2*, or there are numerous cases in the family. The carrier of mutations in *BRCA1/2* genes among women who have developed breast cancer by the age of 35 is about 10%, while for the general population it is about 0.2% [4]. The greater availability of multi-gene sequence analysis by next-generation sequencing (NGS) has improved diagnosis and, consequently, the possibility of using appropriate therapeutic approaches in *BRCA1/2* gene mutation carriers.

Due to the lack of proven efficacy of screening among young women outside of certain high-risk groups, one of the main symptoms of the disease is a breast tumor, i.e., a change that can be felt during self-examination. As a result, cancers in young women are usually found later at higher stages, which reduces the chances of recovery [5]. However, the prognosis of breast cancer patients improves with 90% of patients surviving five years and 84% surviving ten years after cancer diagnosis. There are 63% of patients (47% under the age of 40 and 68% over the age of 65) who are diagnosed with cancer located only in the breast; their five-year survival is about 99%. Long patient survival is inseparably related to the possibility of distant complications of oncological treatment, as well as the effects of long-term hormonal therapy [6].

Planning breast cancer therapy depends primarily on the cancer stage, its biological subtype, familial and genetic burden, the patient’s general condition, and the patient’s willingness. On the other hand, age is not an independent factor determining the use of specific therapeutic methods.

Professional development, starting a family, planning offspring, or pregnancy may sometimes be disrupted by a diagnosis of cancer. Treatment of young breast cancer patients affects their reproductive potential by reducing ovarian reserve. It can lead to reversible or permanent premature menopause, decreased libido, and other symptoms of sex hormone deficiency. This requires that, in addition to oncological treatment, patients are offered genetic counseling, oncofertility, psychological assistance, and sexological counseling [7].

Given the number of *BRCA1/2* gene mutation carriers among young breast cancer patients, but also thanks to growing public awareness, among their healthy family members planning offspring, the possibility of benefiting from preimplantation diagnosis and performing cancer-risk-reduction procedures: RRM (risk-reducing mastectomy), RRSO (risk-reducing salpingo-oophorectomy) significantly increase the chance of a genetically burdened person living a healthy life and giving birth to a child not burdened by the parent’s germline mutation.

However, further education of the public and the medical community is needed regarding pregnancy after cancer, the safe timing of conception after treatment, the course of pregnancy, its surveillance and the health of a child born, and the impact of pregnancy on breast and ovarian cancer induction or recurrence.

The goal of this paper is to show methods and examples of fertility counselling for *BRCA1/2* gene mutation carriers, including both patients already affected by cancer and healthy individuals. Planning to have children is one of the basic elements of care for these patients in the face of the possibility of carrying out cancer risk-reducing treatments and treatments to eliminate the transmission of mutations that increase the risk of cancer in the offspring (using preimplantation genetic testing) (Figure 1).

## 2. *BRCA1/2* Mutation Carriers–Special Considerations Regarding Cancer Risk, Prevention, and Treatment

### 2.1. Mutations in *BRCA1/2* Gene and Cancer Risk

The *BRCA1* gene is located on the long arm of the 17th chromosome, and the *BRCA 2* gene is located on the long arm of the 13th chromosome. Both are tumor suppressor genes, and their proteins are involved in the repair of DNA material by a homologous recombination mechanism [8,9,10].

Mutations in *BRCA1/2* genes are passed from one generation to the next, leaving each child a 50% chance of inheriting the mutation. A germline mutation in *BRCA1/2* genes is associated with an increased risk of breast cancer, ovarian cancer, fallopian tube cancer, primary peritoneal cancer in women, breast cancer, and prostate cancer in men, among others. The risk of breast cancer for *BRCA1* gene mutation carriers by age 80 is 72% and for ovarian cancer 44%, while for *BRCA2* gene mutation carriers it is 69% and 17%, respectively. A sudden increase in the risk of breast cancer is observed in *BRCA1* gene mutation carriers between the ages of 30 and 40, and in *BRCA2* gene mutation carriers between the ages of 40 and 50, after which the risk remains at a similar level until the age of 80. The risk of developing cancer of the other breast is much higher than the general population; in carriers of mutations in the *BRCA1* gene it is 40%, and in the *BRCA2* gene it is 26% [11].

The *BRCA1/2* gene mutation carriers are more frequently diagnosed with tumors of higher histological grade G3. In *BRCA1* gene mutation carriers, the expression of estrogen and progesterone receptors in tumor cell nuclei is observed much less frequently than in the control population (10% vs. 65%), and HER2 gene amplification (3% vs. 15%), respectively. However, in *BRCA2* gene mutation carriers, estrogen and progesterone receptor expression is at a similar level as in the control population (66% vs. 65%; 55% vs. 59%), HER2 receptor status is the same as in BRCA1 gene mutation carriers. The diagnosis rate of a mutation in BRCA1 gene in 30–34-year-old women diagnosed with triple-negative breast cancer is five times higher than the diagnosis of another biological subtype of cancer in a young patient (26.5% vs. 5%) [12]. Genetic counseling is an extremely important part of the diagnostic and therapeutic process of patients with breast cancer. The knowledge of mutation carrier in genes increasing cancer risk provides an opportunity for cancer risk-reducing treatments and in the case of cancer development may determine the choice of surgical treatment (mastectomy vs. breast-conserving surgery, BCT) and systemic treatment (addition of iPARP in complementary therapy for patients with early breast cancer at high risk of recurrence or iPARP vs. chemotherapy for disseminated disease). The *BRCA1/2* gene mutation carriers require family-planning support with the possibility of using PGT-M methods and cascade counseling that includes family members (Figure 2).

### 2.2. RRM and RRSO

Extremely important cancer-risk-reduction procedures include bilateral mastectomy and salpingo-oophorectomy.

RRM reduces the risk of breast cancer by about 90%. Different surgical methods have been used: total mastectomy, skin-sparing mastectomy, nipple-sparing mastectomy. Studies have been conducted on a prospective or retrospective basis with a long-observation period of up to 10 years, and no evidence has been found of an effect of RRM on the prolongation of the overall survival of those who developed the disease [13,14,15,16].

Immediate reconstruction appears to be a safe procedure and can be offered to patients as a part of surgical treatment. Due to the fact that the risk of breast cancer diagnosis during RRM is less than 5%, the sentinel node procedure is not routinely performed. If cancer is diagnosed in the removed gland, deferred surgery in the axillary fossa is indicated.

The CRRM (contralateral risk-reducing mastectomy) reduces the risk of cancer in the other breast, which has been shown in both retrospective and prospective studies with long-observation periods [17,18].

RRSO reduces the risk of ovarian cancer by approximately 80–90% and improves overall survival and cancer-free survival for some patients [19,20,21,22].

Low-certainty evidence suggests that RRSO prolongs ovarian and breast cancer-dependent survival in *BRCA1* gene mutation carriers. The effect of RRSO on cancer-free survival in *BRCA 2* gene mutation carriers is uncertain due to the small number of patients included in the study [23].

The timing of the procedure depends on family history, i.e., the age of onset of cancer in the family, the cancer history of patients, and their desire to have offspring. It is recommended to perform the RRSO at the age of 35–40 years in *BRCA1* gene mutation carriers, and around 40–45 years in *BRCA2* gene mutation carriers because the incidence of ovarian cancer in this group of patients occurs about 8–10 years later than in *BRCA1* gene mutation carriers [24].

### 2.3. PARP Inhibitors

The *BRCA1/2* gene proteins are involved in DNA strand break repair by homologous recombination mechanism. When mutations in *BRCA1/2* genes are present, additional repair mechanisms, including poly (ADP ribose) polymerase (PARP), protect against double-strand DNA damage. The PARP has become an excellent molecular target for therapies (PARP inhibitors) that lead to cell death by a synthetic lethality mechanism [25].

The PARP inhibitors have been used, among others, in the treatment of advanced and early breast cancer. The olaparib and talazoparib have been registered for the treatment of advanced breast cancer in *BRCA* gene mutation carriers. The OlympiAD and EMBRACA trials demonstrated a benefit in terms of increased time to cancer progression (PFS) with a better safety profile and improved quality of life for patients compared to physician’s choice chemotherapy, (7.0 vs. 4.2 months; HR 0.58; 95% CI; 0.43–0.8; *p* < 0.001) and (8.6 vs. 5.6 months; HR 0.54; 95% CI; 0.41–0.71; *p* < 0.001), respectively [26,27].

The benefit of olaparib in complementary treatment was demonstrated in the OlympiA trial. The *BRCA1/2* gene mutation carriers with diagnosed HER2-negative breast cancer at high risk of recurrence after completion of loco-regional therapy and perioperative chemotherapy were treated with olaparib or placebo for one year, either as monotherapy or in combination with hormone therapy and/or bisphosphonates, if recommended. There were 87.5% of patients in the olaparib-treated group and 80.4% in the placebo group who were free of distant metastases after three years of observation, which significantly reduced the risk of recurrence by approximately 40% (HR 0.57; 99.5% CI; 0.39–0.83; *p* < 0.001). There were fewer deaths in the olaparib-treated group, but due to the short observation time, there was no statistical significance [28].

### 2.4. Hormonal Contraception

As already mentioned, family planning in *BRCA1/2* gene mutation carriers is one of the most important components of their care. Childbearing before the recommended age of RRSO or fertility preservation followed by RRSO, as well as the use of assisted reproduction methods with preimplantation genetic testing require cooperation between the *BRCA1/2* mutation carrier, oncologist, geneticist, and reproductive medicine specialist.

One of the elements of family planning is the use of hormonal contraception. In *BRCA1/2* gene mutation carriers it remains a controversial issue. Oral contraception reduces the risk of ovarian cancer, but its effectiveness is lower than the RRSO and, therefore, should not be used solely to reduce the risk of ovarian cancer [29,30,31,32,33].

Data on the effect of oral contraception on breast cancer development are inconclusive. Some studies indicate an increased risk of breast cancer under the influence of oral contraceptive use, while others report just the opposite [30,31,32,34,35].

Some analyses indicate an increased risk of breast cancer among women who started using hormonal contraception before the age of 20 or if diagnosed with cancer before the age of 40 [36,37]. The duration of oral contraceptive use had no effect on breast cancer risk [30,36]. The data analyzed were retrospective studies, and the study populations, especially *BRCA2* gene mutation carriers, were often small. The studies did not evaluate the preparation used or its dose.

The available literature suggests that oral contraceptives reduce the risk of ovarian cancer, but do not reduce the risk enough to safely avoid RRSO and may increase the risk of breast cancer of which patients should be informed.

## 3. Fertility Barriers for *BRCA1/2* Mutation Carriers

### Effects of Oncology Treatment

Fertility problems associated with cancer treatment depend on many factors; the most important are the patient’s age and ovarian reserve, as determined by AMH (anti-Muellerian hormone) levels. The type of anticancer treatment, its doses, duration and routes of administration, doses and irradiation fields, the type of cancer diagnosed, the extent of surgery, and the patient’s general health condition are also extremely important [38,39].

Premature ovarian failure induced by oncological treatment is most often diagnosed on the basis of menstrual absence and low AMH levels. It seems important to determine new markers of ovarian reserve in *BRCA1/2* gene mutation carriers due to described fertility impairment in this group of patients caused by homologous recombination repair disorders and also due to applied oncological treatment, sometimes specific only for patients with genetic burden (PARP inhibitors) [27].

In the treatment of patients with early breast cancer, chemotherapy based on anthracyclines (doxorubicin, epirubicin), cyclophosphamide, taxanes (paclitaxel, docetaxel), platinum derivatives (cisplatin, carboplatin), and capecitabine is used.

Among the drugs with the highest gonadotoxic potential is cyclophosphamide, an alkylating drug that leads to cell death through DNA fragmentation. Its effect is independent of the cell cycle phase. Cyclophosphamide accelerates maturation of primordial follicles and atresia of growing follicles, as well as damages blood vessels and induces inflammation in the stromal, thus leading to premature extinction of ovarian function. The impairment of gonadal function caused by cytostatics is dose and time dependent (Table 1) [40].

Most breast cancers, about 70%, show expression of hormone receptors (estrogen, progesterone) in the nuclei of tumor cells. In patients with hormone receptor positive breast cancer, one of the elements of complementary therapy, which reduces the risk of cancer recurrence, is hormone therapy lasting from five to ten years.

In young patients, aromatase inhibitors which inhibit aromatization of androgens to estrogens mainly in adipose tissue are increasingly used in combination with gonadoliberin analogues. In such patients, it is necessary to check estradiol and FSH levels regularly and if gonadoliberin analogues are not effective, to change hormone therapy to tamoxifen in monotherapy until physiological menopause is reached and then change to IA.

Another therapeutic option is to include tamoxifen in monotherapy or more commonly in combination with a gonadoliberin analogue. The intensity and duration of hormonal treatment depends on, among others, the patient’s age at diagnosis, menopausal status, histologic malignancy and tumor grade, and the level of hormone receptor expression [7].

Hormone therapy does not directly affect the ovarian reserve. However, the recommended long duration of therapy postpones the moment of pregnancy. Tamoxifen used after perioperative chemotherapy increases the risk of menstrual suppression but does not affect AMH levels and, thus, does not impair female reproductive function [41,42]. Among the most aggressive biological subtypes is HER2-positive cancer, which is diagnosed in approximately 15% of patients with early breast cancer.

Preoperative treatment involves two antibodies directed against two different epitopes of the HER2 receptor, pertuzumab and trastuzumab, in combination with chemotherapy. Achieving a complete pathological response, i.e., absence of tumor cells in the surgical specimen after previous systemic treatment, improves the prognosis of patients. In contrast, residual disease requires intensification of adjuvant therapy in which case the immunoconjugate trastuzumab-emtansine is used instead of trastuzumab [43]. So far, no gonadotoxic effect of any of these drugs directed against the HER2 receptor has been proven [38,39].

In the OlympiA trial, the efficacy of iPARP–olaparib therapy was proven in *BRCA1/2* gene mutation carriers with diagnosed breast cancer at high risk of recurrence. Determining the effect of olaparib on fertility in *BRCA1/2* gene mutation carriers will require several more years of intensive research. Previous studies in mice indicate that olaparib administered in monotherapy reduced the number of primordial follicles by approximately 36% without affecting the number of remaining follicles or AMH levels. However, olaparib used after prior chemotherapy did not potentiate its follicle-destroying effects [28].

## 4. Fertility Counseling in *BRCA1/2* Gene Mutation Carriers

### 4.1. Fertility Preservation Techniques

The procedure of fertility preservation by ovarian stimulation and oocyte retrieval prolongs the time to oncological treatment by about 8–10 days. However, it has no effect on both invasive disease-free survival (IDFS) and overall survival (OS) of women with BRCA1 and two mutations.

It is now accepted that controlled ovarian stimulation can be initiated on any day of the cycle, reducing the time to oncological treatment. Oocyte or embryo freezing is the basic method of fertility preservation in breast cancer patients, including *BRCA1/2* gene mutation carriers. The number of frozen oocytes or embryos depends on the patient’s age and basal ovarian reserve. There was no influence of neoplasia on the response to stimulation. However, premature ovarian failure was observed in *BRCA1* gene mutation carriers and consequently, decreased ovarian reserve and poorer response to stimulation, irrespective of the neoplasia. In women with a low ovarian reserve and not requiring rapid oncological treatment, it is possible to perform two stimulations, one after the other, which increases the chance of greater procedure efficacy [44]. IVF in *BRCA1/2* gene mutation carriers allows for the performance of preimplantation genetic testing as well as to avoid passing the genetic load to the offspring.

In 2003, the Ethics Taskforce ESHRE (European Society of Human Reproduction and Embryology) accepted preimplantation genetic testing for single-gene diseases in case of hereditary breast ovarian cancer (HBOC) diagnosis [45].

In 2008, Jasper et al. first reported a case of a child born after PGT-M by a *BRCA1* gene mutation carrier. Derks-Smeets et al. presented a paper on 70 couples who underwent PGT-M due to the burden of carrying mutations in the *BRCA1/2* genes [46,47]. In patients with breast cancer, especially hormone-receptor positive breast cancer, hormonal stimulation is still controversial. The use of letrozole during stimulation leads to a decrease in estradiol levels by more than 50% compared to patients undergoing elective stimulation for fertility preservation, mainly due to age. The effect of aromatase inhibitor on the method effectiveness is inconclusive. The number of mature MII oocytes is lower than with other stimulation regimens. The reason for the lower number of mature oocytes as a result of stimulation with letrozole cannot be unambiguously stated. Further analysis of the tumor subtype or genetic load (mut. *BRCA1/2*), among others, is necessary. No influence of hormonal stimulation and IVF on the increased risk of tumor recurrence or on the occurrence of congenital defects in children has been observed [48,49,50,51].

Among the most important factors determining the procedure’s efficacy is age. In a study conducted by von Wolff et al., among women with various cancers, the number of collected oocytes in patients < 26 years of age was 15.4 ± 8.8, in patients 26–30 years of age it was 13.1 ± 8.5, in patients 31–35 years of age–12.2 ± 7.7, and in patients 36–40 years of age–9.9 ± 8.0. No differences were observed depending on the type of cancer, but apart from age, previous ovarian surgeries which may decrease the initial ovarian reserve were also relevant [52].

Oncological treatment for breast cancer reduces the chance of natural pregnancy by 60%. In a study by Cobo et al., 7.4% of oncology patients who preserved fertility with oocyte or embryo freezing reported to a reproductive medicine specialist for their use. The low percentage of patients may have been due to a number of reasons, including young age at breast cancer diagnosis; 70% of patients were younger than 35 years old, increasing their chances of becoming pregnant naturally after completion of cancer treatment, long duration of therapy, or cancer recurrence, and focus on causal treatment. In the subgroup of young women under 35 years of age, the survival of oocytes after unfreezing was significantly different (81.2% vs. 91.4%); the cumulative clinical pregnancy rate (42.8% vs. 65.9%), the cumulative ongoing pregnancy rate (35.7% vs. 57.7%), and the cumulative live birth rate (42.1% vs. 68.8%) were lower in the subgroup of women treated with oncology than in women after elective fertility preservation. Such surprising differences may be explained by the fact that in non-oncology patients, only top-quality cells were frozen, whereas in oncology patients, the freezing procedure was performed, regardless of cell quality, in an attempt to increase the sense of purpose in the lives of those treated. The use of anticancer treatment damages n10.1200/JCO.19.02399ot only the ovaries, directly reducing ovarian reserve, but also affects the endometrium, reducing the chance of embryo implantation [53].

According to Gellert et al. ovarian tissue freezing and transplantation is performed in 21 countries, with 360 transplantation procedures performed in 318 women to date. Cancer is the main reason for freezing ovarian tissue to preserve fertility. In 95% of women, return of ovarian hormonal function was observed within four to nine months. Of the 318 women, 170 made the decision to have a baby. In 95 women, 131 pregnancies were confirmed of which 69 delivered 87 healthy babies. Half the children born were conceived naturally, and one case of congenital malformation in the child was diagnosed, which was related to genetic burden [54].

Ovarian tissue freezing should be used in *BRCA1/2* gene mutation carriers with particular caution due to the possibility of ovarian transplantation with developing cancer. In an analysis conducted by Gellert et al., nine women with transplanted ovarian tissue were diagnosed with cancer. However, no direct relationship with the procedure was found [55]. Although ESMO recommends this procedure only for women under 37 years of age, there is no logic to this approach (ESMO). The sensibility of the procedure depends on ovarian reserve, and this only correlates with age and is not identical in every woman of the same age. Moreover, if no other procedure is possible, why not use this one even with relatively low ovarian reserve? After all, there are pregnancies even at extremely low AMH levels, and in these cases, the problem of low efficacy is in both older and younger women [56].

Therefore, the procedure of ovarian tissue freezing should be limited to patients carrying mutations in *BRCA1/2* genes in whom hormonal stimulation cannot be performed or who require immediate oncological treatment, mainly due to the risk of autotransplantation of ovarian tissue with cancer cells. It has the advantage of possible ovarian tissue collection during cytotoxic treatment. However, the intensity of treatment may affect the obtained material. The effect of different therapies used prior to ovarian tissue retrieval on a woman’s subsequent reproductive capacity after re-transplantation is unknown.

In *BRCA1/2* gene mutation carriers, orthotopic transplantation of ovarian tissue is recommended to allow removal of previously left ovaries and fallopian tubes (RRSO) in a single procedure after reproduction. In the case of performing RRSO and then freezing the ovarian tissue to preserve fertility, heterotopic autotransplantation into the forearm or abdominal region also seems reasonable, especially in patients after radiotherapy of the pelvic region or if reoperation may be difficult due to, among others, adhesions. This facilitates cycle monitoring and subsequent oocyte retrieval. A case of natural pregnancy after heterotopic ovarian autotransplantation was also described. A 32-year-old patient with Hodgkin’s lymphoma was diagnosed with POF after high-dose chemotherapy before bone marrow transplantation. After 2.5 years and following transplantation of ovarian tissue into the suprapubic region, the patient became pregnant twice and delivered a healthy daughter at 40 weeks gestation. It is supposed that the transplanted ovarian tissue can stimulate follicle growth in the previously left ovary [57,58].

Prior to transplantation, ovarian tissue should be evaluated by available methods (immunohistochemistry, molecular markers, xenografting) to minimize the risk of transplanting tumor cells along with the ovarian fragment [38].

Patients who do not agree with fertility preservation with the presented methods may be suggested to include GnRHa during oncological treatment but at least one week before starting the therapy. This method can also be used as an additional option for securing ovarian function, which reduces the risk of premature extinction of ovarian function and increases the number of pregnancies without affecting the safety of cancer therapy, regardless of the hormone receptor status of the cancer. Data on breast cancer patients carrying mutations in *BRCA1/2* genes are considerably limited, but case series descriptions indicate that GnRHa is effective in these patients during chemotherapy. Such patients may be more interested in fertility preservation procedures with the possibility of subsequent preimplantation diagnosis or RRSO immediately after completion of cancer treatment. Therefore, the use of GnRH during chemotherapy will be irrelevant to these patients [53,59].

### 4.2. Preimplantation Genetic Testing–Types and Applications

Preimplantation testing has been used since 1990. It became common practice with data that pointed to genetic abnormalities in embryos as a major cause of low human fertility and a growing problem as women try to get pregnant. Embryo aneuploidy is a major cause of both poor embryonic development and embryonic arrest, lack of implantation, and early miscarriage of pregnancies. It is now believed that aneuploidies account for over 80% of the causes of the above problems. That is why preimplantation genetic testing for aneuploidies (PGT-A) is so widely used. In the United States already about 30% of all IVF procedures are performed with the use of these diagnostics.

Decreased ovarian reserve in women with *BRCA1/2* mutations causes premature cessation of ovarian function and with it a growing risk of aneuploidy. Therefore, this group of women has particular indications for preimplantation diagnosis of aneuploidy. This applies in the cases of trying to get pregnant before the disease, both in the case of prophylactic fertility preservation or emergency procedure at the time of diagnosis and before the implementation of oncological treatment and in the course of trying to get pregnant after the cancer recovery.

In the case of mutation carriers, it is also possible to prevent transmission to the offspring. Another type of preimplantation diagnosis is preimplantation genetic testing for monogenic diseases (PGT-M). This method allows us to determine which embryos will carry a mutation. Under optimal conditions, it is possible to implant only embryos that do not carry this mutation, thus eliminating the ongoing misery in the family. For this reason, it is believed that every woman who knows about her *BRCA1/2* gene mutation, including the fertile one, should consider IVF with the exclusion of carrying the mutation in her offspring. In addition to the undoubted impact of this procedure on the health of the offspring, the cost-effectiveness of the conception through IVF/PGT-M for families with high *BRCA*-related morbidity and mortality compared to natural conception has also been demonstrated in a recent study by Michan et al. [60].

Preimplantation testing continues to develop. Currently, it is possible to diagnose many genetic diseases, including those of a simultaneous multigene etiology. Such diagnostics is called preimplantation genetic testing for polygenic diseases (PGT-P). However, its dissemination will cause the compulsion to make ethically difficult decisions about the choice of embryo for transfer depending on the expressed percentage of predisposition to various diseases, such as various cancers, cardiovascular diseases, or diabetes [61,62].

## 5. Pregnancy in *BRCA1/2* Mutation Carriers

Pregnancy in *BRCA1/2* gene mutation carriers requires appropriate planning. First, family history and recommended RRM and RRSO should be considered. Some studies indicate worse ovarian reserve and poorer response to stimulation compared to the control group, which further encourages earlier attempts to have a child.

The chance of pregnancy for women who have had breast cancer is 60% lower than for healthy women [56].

Studies show that less than 10% of women choose to become pregnant after a breast cancer diagnosis but almost twice as many in the population of *BRCA1/2* gene mutation carriers (19%). This may be related to the very young age of cancer onset, the need to schedule cancer-reducing treatments, and the rarer incidence of hormone receptor positive breast cancers that require long-term complementary therapy. The risk of miscarriage, preterm birth, or congenital malformations in children of *BRCA1/2* gene mutation carriers born after oncological treatment was at a similar level as in the general population, respectively (9.2% vs. 11%), (10.3% vs. 17%), (1.8% vs. 3%) [63,64].

In our largest retrospective analysis to date, which included 1252 breast cancer patients (811 *BRCA1* gene mutation carriers, 430 *BRCA2* gene mutation carriers, and 11 *BRCA1 and 2* gene mutation carriers), 195 patients became pregnant at least once and 150 of them gave birth for a total of 170 children. The median observation time was 8.3 years, and no differences in time to cancer recurrence (DFS) (HR 0.87; 95% CI, 0.61–1.23, *p* = 0.41) and overall survival (OS) (HR 0.88, 95% CI, 0.5–1.56, *p* = 0.66) were observed between the subgroup of patients who became pregnant and the subgroup of patients who did not become pregnant. In subgroup analysis, pregnancy in *BRCA1* gene mutation carriers had a protective effect, while in *BRCA2* gene mutation carriers it had a negative effect on patient prognosis. However, the data presented should be analyzed with caution due to the small number of patients with a *BRCA2* mutation (*n* = 44) who were included in the study. Data on the effect of hormone receptor status on patient prognosis are inconclusive and need to be confirmed in subsequent studies [65].

Given that most breast cancers express hormone receptors, some clinicians may have concerns about planning offspring by breast cancer patients. In 2013, preliminary results from a retrospective analysis conducted by Azim Jr. et al. on the impact of pregnancy on patient prognosis according to hormone receptor status were published. After a median observation period of 4.7 years following pregnancy, there was no difference in time to cancer progression (DFS), regardless of pregnancy or no pregnancy in either the hormone receptor positive group (HR = 0.91; 95% CI, 0.67 to 1.24, *p* = 0.55) or hormone receptor negative (HR = 0.75; 95% CI, 0.51 to 1.08, *p* = 0.12) and longer overall survival (OS) in patients who became pregnant (HR = 0.72; 95% CI, 0.54 to 0.97, *p* = 0.03), regardless of estrogen receptor status (*p* = 0.11) [64]. As patients with hormone receptor positive cancers are more likely to experience late recurrence of breast cancer than other biologic subtypes, longer observation time for patients who became pregnant was necessary to assess the impact of pregnancy on oncologic safety. In 2018, Lambertini et al. published updated data, following a median observation period of 7.3 years after pregnancy. Again, no difference in time to cancer recurrence (DFS) was observed between patients who became pregnant or did not become pregnant, regardless of whether they were diagnosed with hormone receptor positive (hazard ratio [HR] = 0.94, 95% confidence interval [CI] = 0.70 to 1.26, *p* = 0.68) or hormone receptor negative (HR = 0.75, 95% CI = 0.53 to 1.06, *p* = 0.10) cancer. There was no difference in OS in the group diagnosed with estrogen-dependent cancer (HR = 0.84, 95% CI = 0.60 to 1.18, *p* = 0.32); there was an OS benefit for patients who became pregnant with previously diagnosed hormone receptor positive cancer (HR = 0.57, 95% CI = 0.36 to 0.90, *p* = 0.01). The data collected refers to patients diagnosed with breast cancer diagnosed before 2007; *BRCA1/2* gene mutation carriers were not included in the analysis, which may be due to diagnostic limitations at that time [66].

Although triple negative cancers are diagnosed more frequently in *BRCA1/2* gene mutation carriers than in the population of genetically non-burdened patients of the same age, the vast majority of cancers show expression of hormone receptors. Complementary hormone therapy should be given for 5–10 years, which postpones the time when pregnancy will be possible or completely precludes further having offspring. Studies have shown that the desire to have offspring was a reason for discontinuing or not starting hormone therapy for some patients of reproductive age [67]. This prompted the researchers to conduct a study among patients diagnosed with hormone receptor positive cancer who were given a maximum two-year break of 18–30 months after starting complementary hormone therapy to attempt pregnancy, childbirth, and breastfeeding. After this time, hormone therapy was reinitiated for the remaining preplanned duration of therapy. During the five years of study recruitment, 518 patients randomized at 116 centers, in 20 countries on four continents were included. The median age at study recruitment was 37 years; 74.9% of women had no previous offspring. Fertility preservation was used in 51.5% of women. In 93.2% of patients, stage I or II breast cancer was diagnosed, 66% without lymph node involvement, and 61.9% received preoperative chemotherapy. The most common type of complementary hormone therapy was tamoxifen (41.8%), followed by tamoxifen in combination with ovarian function suppression (OFS) (35.4%) and aromatase inhibitors in combination with OFS (15.9%). In the second half of recruitment (after July 2017) and the announcement of the SOFT/TEXT trial results, the proportion of hormone therapy used reversed in favor of IA.

Only thirty-eight *BRCA1/2* gene mutation carriers were included in the study (18 in *BRCA1* gene, 20 in *BRCA2* gene), 226 patients did not have *BRCA1/2* gene mutations, 236 patients did not undergo genetic diagnosis, and test results were unavailable in 17 patients. Given the relatively high ages of the patients included in the study (median age was 37 years) and the fact that most of them (74.9%) had no offspring or 20.9% had one child, the study may have been particularly attractive to patients. Considering the relatively low cancer stage, discontinuation of hormone therapy was a safe procedure from the oncologist’s point of view because of the low risk of cancer recurrence [68,69]. The presented data indicate that pregnancy after completion of oncological treatment for breast cancer does not affect the prognosis of patients, regardless of *BRCA1/2* gene mutation status and hormone receptor expression and does not affect the well-being of the children born. Appropriately planned, prior to RRSO, it provides an opportunity for family life even after cancer. Thanks to the increasing availability of PGT-M, it enables avoidance of pathogenic mutation inheritance by the offspring.

## 6. Conclusions

Both healthy *BRCA1/2* gene mutation carriers and women genetically burdened with breast cancer require consultation with a reproductive medicine specialist for individual planning of offspring. The cooperation of specialists in various fields (reproductive medicine, gynecology, genetics, oncology), at every stage of patient care, allows one to maintain oncological safety and enjoy family life.

## Figures and Tables

**Figure 1 jcm-11-03996-f001:**
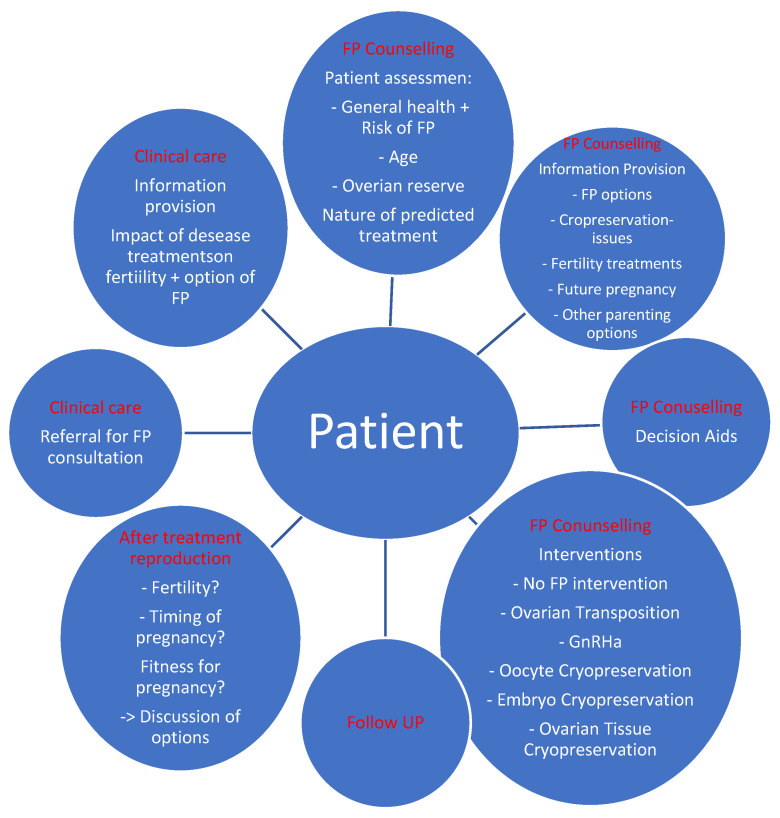
Model of care for patients eligible for fertility preservation.

**Figure 2 jcm-11-03996-f002:**
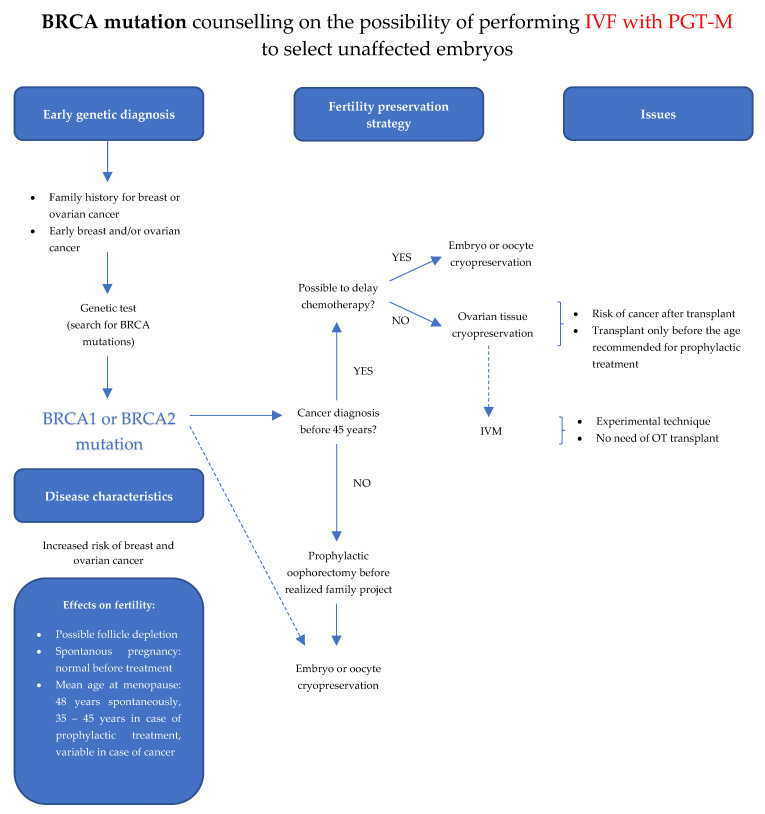
Fertility preservation strategy in patients with BRCA genes mutations.

**Table 1 jcm-11-03996-t001:** Risk of treatment-related amenorrhea in female patients.

Degree of Risk	Regiment	Comments
High risk (>80%)	6 cycles of CMF, CEF, CAF/TAC in women of ≥40 years	Significant decline in AMH level after treatment
Intermediate risk (20–80%)	6 cycles of CMF, CEF, CAF/TAC in women of 30–39 years 4 cycles of AC in women of ≥40 years 4 cycles of AC/EC → taxane 4 cycles od cc (F)EC → taxane	Significant decline in AMH level after treatment
Low risk (<20%)	6 cycles of CMF, CEF, CAF/TAC in women of <30 years 4 cycles of AC in women of <40 years Antimetabolites (methotrexate, fluorouracil) Vinca alkaloids Bevacizumab	Significant decline in AMH level after treatment
Unknown risk	Platinum- and taxane-based ChT Most targeted therapies Immunotherapy	

AC—doxorubicin, cyclophosphamide; AMH—anti-Mullerian hormone; CAF—cyclophosphamide, doxorubicin, 5-fluorouracil; CEF—cyclophosphamide, epirubicin, 5-fluorouracil; Cht—chemotherapy; CMF—cyclophosphamide, methotrexate, 5-fluorouracil; EC—epirubicin, cyclophosphamide; FEC—5-fluorouracil, epirubicin, cyclophosphamide; TAC—docetaxel, doxorubicin, cyclophosphamide. Adapted with permission from Ref. [38].

## Data Availability

Not applicable.

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
