# Peer review of "Fertility Counseling in BRCA1/2-Mutated Women with Breast Cancer and Healthy Individuals"

_jcm, 2022, doi:10.3390/jcm11143996_

Round 1

Reviewer 1 Report

Kufel-Grabowska et al wrote a review about the impacts of BRCA1/2 mutations in both cancer patients and healthy patients for fertility counseling patients. The review is extensive and cites even contraception counseling and methods such as fertility preservation and preimplantation genetic testing in order to help carriers and patients with these mutations. The work shows lots of information that are important for physicians or other health professionals who deal with patients with these mutations. The language and table are good. It is a little bit difficult to read the texts in Figure 2. It should improve the quality of the figure.

Since it is a review article, I don't understand why the authors use the numeration for 1- Introduction, 2- Material and Methods, 3- Results... If I understood the dynamics of the review it should be something like this: 1- Introduction, 2- MUTATIONS IN BRCA 1/2 GENES AND CANCER RISK 97 ,3- ...

In addition, I felt that this review was written in the same way of an introduction of a Ph.D. thesis, with many topics that could be condensed into one, I recommend to the authors to see examples of review publications at PCM journal to see how is the structure of a review.

For example: The goal is to show methods and examples of fertility counselling for carriers and cancer patients with BRCA1/BRCA2. I can understand the second topic after the introduction, which describes what are BRCA1/BRCA2 mutations and how they are important for this topic. But why create a topic for the types of surgeries and treatments for breast cancer (like parp inhibitors)? I don't understand it as a part of the text flow. 

The topics must be connected, it should be something like that(it doesn't need to be this exact topic name): Introduction -> BRCA1/2 Mutations -> Fertility barriers for these patients/carriers --> Fertility counseling methods to assist BRCA1/BRCA2 mutation carriers --> Fertility counseling for healthy carriers vs cancer patients.

It is even not clear what are the methods used for healthy carriers vs cancer patients. Mostly the review needs a better re-organization to improve the flow to achieve the goal.

There are some minor questions about the review as well:

  • In line 79, RRM is defined as Risk Reducing Mastectomy, but in line 135 RRM is re-defined as Bilateral mastectomy. Which one is correct? And why define this acronym too many times?
  • In lines 207 and 208 there are two different headings for the same topic. Which one is correct? It looks like that it is talking about treatment effects on fertility, correct?

Author Response

Dear Reviewer, 

Thank you for your useful comments. We re-organized our manuscript according your suggestion. Now our numeration of sections should be more clear for readers.

Reviewer 2 Report

Dear Authors,

Thank you for giving me the opportunity to review this manuscript.

I believe that the aim of this paper is really interesting and worthy of appreciation; the scientific contribution of the review can be really important since it concerns a topic of great oncological interest.

I think the relevance of the review topic was covered, with the appropriateness of references.

I really appreciate the focus “model of care  for patients elegible for fertility preservation”.

However, I don’t understand the choice (it’s a choice?) not to include the section “methods” to explain “Eligibility criteria”, “Search strategy”, “Selection process”, “Data collection process”, etc.

I attached the PRISMA 2020 Checklist to improve your work…I think it would be useful for you.

I suggest adding a flow diagram (I also attached an example) to explain the different stages of the data collection.

Thank you

Author Response

Thank you for your useful comments. We decided to not add materials and methods section, because our manuscript is not a systematic review but describes specific situations with BRCA mutation patients, and they relate to their reproductive potential. 

Round 2

Reviewer 1 Report

The authors fulfilled all my previous comments.

There isn't any additional comment 

Reviewer 2 Report

Dear Authors,

I see that you have worked hard and the manuscript has improved since the first time. I know your manuscript is not a systematic review, but my suggestion is to complete the material and methods section in this type of narrative review as well.  You can add keywords and years of research, for example...